Body fluids and muscle changes in trail runners of various distances

Cebrián-Ponce Álex 1
http://orcid.org/0000-0001-8779-8745 Marini Elisabetta 2
Stagi Silvia 2
http://orcid.org/0000-0003-0517-9110 Castizo-Olier Jorge 1 3
Carrasco-Marginet Marta 1 mcarrascom@gencat.cat
http://orcid.org/0000-0003-1210-6171 Garnacho-Castaño Manuel Vicente 3 4
Noriega Zeasseska 1
Espasa-Labrador Javier 1
Irurtia Alfredo 1
1 INEFC-Barcelona Sports Sciences Research Group, National Institute of Physical Education of Catalonia (INEFC). University of Barcelona (UB) , Barcelona , Spain
2 Department of Life and Environmental Sciences. Neuroscience and Anthropology Section, University of Cagliari , Cagliari , Italy
3 DAFNiS Research Group (Pain, Physical Activity, Nutrition and Health), Campus Docent Sant Joan de Déu, University of Barcelona , Barcelona , Spain
4 Faculty of Health Sciences, Valencian International University (VIU) , Valencia , Spain
Jimenez Manuel
Electronic publication date: 2023 Dec 1
Publication date: 2023
Volume: 11
Electronic Location ID: e16563
Received 2023 Sep 20; Accepted 2023 Nov 12
Copyright: © 2023 Cebrián-Ponce et al.
Copyright year: 2023
Copyright holder: Cebrián-Ponce et al.
License: This is an open access article distributed under the terms of the Creative Commons Attribution License, which permits unrestricted use, distribution, reproduction and adaptation in any medium and for any purpose provided that it is properly attributed. For attribution, the original author(s), title, publication source (PeerJ) and either DOI or URL of the article must be cited.
License URL: https://creativecommons.org/licenses/by/4.0/

Keywords: Body cell mass, Hydration, Phase angle, BIVA, Sport, Running

Funding: National Institute of Physical Education of Catalonia (INEFC) This work was supported by the National Institute of Physical Education of Catalonia (INEFC) of the Generalitat de Catalunya. The funders had no role in study design, data collection and analysis, decision to publish, or preparation of the manuscript.

==============================
Background

This study aims to investigate body fluids and muscle changes evoked by different trail races using anthropometric, bioelectrical, and creatine kinase (CK) measurements.

Methods

A total of 92 subjects (55 men, 37 women) participating in three different races of 14, 35, and 52 km were evaluated before (PRE) and after (POST) the races. Classic bioelectrical impedance vector analysis was applied at the whole-body level (WB-BIVA). Additionally, muscle-localized bioelectrical assessments (ML-BIVA) were performed in a subgroup of 11 men (in the quadriceps, hamstrings, and calves). PRE-POST differences and correlations between bioelectrical values and CK, running time and race distance were tested.

Results

Changes in whole-body vectors and phase angles disclosed an inclination towards dehydration among men in the 14, 35, and 52 km groups (p < 0.001), as well as among women in the 35 and 52 km groups (p < 0.001). PRE Z/H was negatively correlated with running time in the 35 km men group and 14 km women group (r = −0.377, p = 0.048; r = −0.751, p = 0.001; respectively). POST Z/H was negatively correlated with running time in the 14 km women group (r = −0.593, p = 0.02). CK was positively correlated with distance in men and women (p < 0.001) and negatively correlated with reactance and vector length in the 14 km men group (p < 0.05). ML-BIVA echoed the same tendency as the WB-BIVA in the 35 and 52 km runners, with the most notable changes occurring in the calves (p < 0.001).

Conclusions

WB-BIVA and CK measurements underscored a conspicuous trend towards post-race dehydration and muscle damage, displaying a weak association with performance. Notably, ML-BIVA detected substantial alterations primarily in the calves. The study underscores the utility of BIVA as a technique to assess athlete’s body composition changes.

Introduction

In recent years there has been a growing interest in natural environment running, and trail running (TR) has become an increasingly popular sport (Carmona et al., 2019). The International Trail Running Association defines TR as “races which take place in a natural environment, with the minimum possible of paved roads (20% maximum)” (ITRA, 2021).

Many physiological changes occur during and at the end of the races (Baiget et al., 2018; Carmona et al., 2019; Koral et al., 2022; Pradas et al., 2021; Roca, 2019). Dehydration processes and neuromuscular fatigue, especially occurring in the lower limb (Roca, 2019), have been observed (Baiget et al., 2018; Carmona et al., 2019; Koral et al., 2022; Pradas et al., 2021; Roca, 2019). The hydration status prior to competition and strategies for maintaining euhydration during it are important, given their potential to influence athletic performance (Gatterer, 2021). The lack of body water increases the physiological strains and perceived effort in aerobic exercises (Sawka et al., 2007). In fact, there is general agreement that a ≥2% body mass (BM) loss affects aerobic and cognitive performance (Casa, Clarkson & Roberts, 2005). Conversely, excessive fluid intake can lead to increased BM, causing a weight penalty and probably affecting the weight-bearing activities (Gatterer et al., 2011; Noakes, 2007; Sawka et al., 2007). Therefore, for achieving optimal endurance performance, attention should be paid to proper hydration status (Gatterer, 2021), that together with good neuromuscular conditions, represent useful predictors of running performance (Baiget et al., 2018; Pastor et al., 2022).

Neuromuscular fatigue is often accompanied by increases in plasma biomarkers of muscle damage, such as creatine kinase (CK) (Brancaccio, Lippi & Maffulli, 2010). When the muscle tissue cannot stand the intensity of the exercise overreaching its limit, CK is released into the interstitial space and transported into the bloodstream through the lymphatic system (Brancaccio, Lippi & Maffulli, 2010). CK has been widely studied in TR (Carmona et al., 2019; Pradas et al., 2021), revealing large increases at the end of the races. CK elevation is usually lower in trained individuals (Carmona et al., 2019), because their basal level and their exercise tolerance are usually higher (Kim & Lee, 2015; Koutedakis et al., 1993). Moreover, a positive correlation between race distance and CK levels has been observed (Temesi et al., 2021).

Various methods for analyzing hydration status exist within the sports field, like the plasma analysis or dilution techniques (Roca, 2019). While these methods offer valuable insights, some are impractical, costly, or invasive. Monitoring changes in BM, as well as food and fluid intake, constitute additional informative methods of the hydration status (Lukaski, 2017). A noteworthy addition to this array of methods is the bioelectrical impedance vector analysis (BIVA), which offers a comprehensive approach to assessing body hydration and cellular status (Campa et al., 2022). The classic BIVA approach works with the same basis of bioelectrical impedance analysis (BIA), but instead of using regression equations to obtain quantitative data, it relies on the analysis of raw bioelectrical values (resistance, R; reactance, Xc) standardized by conductor length. The derived impedance (Z) and phase angle (PhA) represent vector length and vector direction, respectively (Piccoli et al., 1994). According to classic BIVA, vector length is inversely related to total body water (TBW) (Campa et al., 2022), whereas vector direction serves as an indicator of cellular health and cell membrane integrity and is inversely related to extracellular/intracellular water (ECW/ICW) ratio (Marini et al., 2020). BIVA can be performed through different protocols depending on the distribution of the electrodes (Campa et al., 2022; Stagi et al., 2021a), including whole-body BIVA (WB-BIVA) for assessing the composition of the entire body, and muscle-localized BIVA (ML-BIVA), for assessing the composition of individual muscles or muscle segments. BIVA has been studied in different applications, such as in the clinical field (Norman et al., 2012), or in body image and body composition (Stagi et al., 2021b). In the sports field, some studies have employed both WB-BIVA and ML-BIVA to assess fluid fluctuations following training sessions or competitions (Campa et al., 2022; Castizo-Olier et al., 2018b; Cebrián-Ponce et al., 2021, 2023; Nescolarde et al., 2023). Notably, Castizo-Olier et al. (2018a) reported that classic whole-body vector migration remained consistent with fluid loss induced in an ultra-endurance triathlon event, even 48 h after the race finished. Nescolarde et al. (2020) established a connection between classic whole-body vector migration and certain serum and urine biomarkers post-marathon, providing insights into hydration status and renal function. Furthermore, a recent study revealed a positive correlation between resistance (BIVA) and muscle damage following a high-intensity hockey training (Cebrián-Ponce et al., 2022). To the best of our knowledge, this study represents the first attempt to employ BIVA in the context of TR, offering valuable insights into the physiological responses elicited by these distinctive races.

This study aimed to analyze body fluids and muscle changes evoked by different trail running race profiles (14, 35 and 52 km) by using anthropometric, bioelectrical (whole-body and muscle-localized) and CK assessments.

Materials and Methods

Subjects

As shown in Table 1, this observational and descriptive study involved 92 trail runners of both sexes (55 men and 37 women). Each participant engaged in one of the three races constituting the Volta a la Cerdanya Ultrafons® 2013 evet (Fig. 1): (a) a 14 km race with 489 m of elevation gain; (b) a 35 km race with 1,600 m of elevation gain; (c) an 85 km race with 3,800 m of elevation gain. However, due to the adverse weather conditions during the competition, the 85 km race had to be shortened to 52 km, with 2,300 m of positive elevation gain. Participants were surveyed about their weekly training volume, revealing the typical heterogeneity among trail runners. Runners selected their race based on their preparedness and physical fitness. The inclusion criteria consisted of: (a) participants aged 18 and older; (b) absence of injuries or clinical conditions at the time of the study.

Table 1 Characteristics of the study participants and performance.

		Men	Women	
Age (years)	
	14 km	35.8 ± 8.4	35.1 ± 11.4	
	35 km	37.2 ± 7.8	35.5 ± 6.7	
	52 km	36.7 ± 4.8	34.3 ± 3.8	
Height (cm)	
	14 km	176.8 ± 7.5	165.5 ± 6.4	
	35 km	176.1 ± 6.4	164.2 ± 6.6	
	52 km	175.9 ± 8.0	163.9 ± 3.7	
Race time (min)	
	14 km	103.4 ± 20.5	123.2 ± 13.4	
	35 km	226.3 ± 39.7	282.2 ± 52.4	
	52 km	408.6 ± 75.8	464.8 ± 90.2	

Figure 1 Elevation profile of the races.

Blue: 14 km; Green: 32 km; Red: 85 km, shortened at 52 km (vertical red line).

The study was performed following the Helsinki Declaration Statement. All runners voluntarily participated and delivered written informed consent prior to their participation. The research was previously approved by the Ethics Committee of the Catalan Sports Council (0099 S/690/2013).

Procedures

Anthropometric, bioelectrical and hematological measurements were conducted in the morning under fasting conditions before (PRE) and at the end of the race (POST). Blood samples were collected from an antecubital vein by a physician, following the same protocol as previous studies (Carmona et al., 2019). CK analysis were performed in an Advia 2400 automatic device (Siemens Medical Solutions Diagnostics, Tarrytown, NY, USA). Out of the 92 subjects, blood samples were obtained from all of them at PRE, but could not be obtained from seven women at POST.

Anthropometry

The anthropometric measurements adhered to the standard criteria established by the International Society for the Advancement of Kinanthropometry (ISAK) (Stewart et al., 2011), using the same equipment as in Cebrián-Ponce et al. (2023). The anthropometric tape was also utilized to assess perimeters (mid-thigh and calf maximum) and length segments (quadriceps, hamstrings, and calves).

Bioelectrical impedance analysis

The impedance measurements were collected using the BIA device (BIA 101 Anniversary; Akern, Florence, Italy), applying a constant alternating sinusoidal electric current at a frequency of 50 kHz.

WB-BIVA and ML-BIVA (for quadriceps, hamstrings, and calves) data were obtained following the detailed procedures outlined in Cebrián-Ponce et al. (2023) and visually depicted in Fig. 2. WB-BIVA is a conventional method widely used in various scientific and clinical settings to assess overall body composition. On the other hand, ML-BIVA is a more recent approach that allows us to examine specific muscle segments, providing valuable insights into localized muscle changes. This combined approach enhances our ability to comprehensively understand body composition alterations.

Figure 2 Electrode placement for bioelectrical impedance measurements in the whole-body (A), quadriceps (B), hamstrings (C), and calves (D).

Z was computed as √(R² + Xc²), and PhA was determined using the formula tan-1 (Xc/R · 180°/π). The parameters R, Xc, and Z were adjusted with respect to height (R/H, Xc/H, Z/H) in the case of WB-BIVA, and with respect to segment length (R/L, Xc/L, Z/L) for ML-BIVA. The parameters Z/H and PhA were jointly analyzed to assess changes in TBW and ECW/ICW ratio (Campa et al., 2022; Marini et al., 2020), using athlete-specific prediction models proposed by Matias et al. (2016) based on H, BM, R, Xc, and gender.

Statistical analysis

Descriptive data are presented as mean ± standard deviation. After evaluating the normality of the distribution for each variable using the Shapiro-Wilks test, differences between PRE and POST values were assessed using the paired Student’s t-test for parametric distribution and the Wilcoxon signed-rank test for non-parametric distribution. The magnitude of changes in all values was expressed as delta percent values (∆%). Pearson’s correlation coefficient was employed to identify potential associations between changes in bioelectrical parameters and CK delta percent values, as well as between PRE and POST bioelectrical parameters and running time. Spearman rank-order correlation coefficient was applied to examine the relationships between race distance and CK changes. Resistance-reactance paired graphs and paired one-sample Hotelling’s T2 test were used to compare bioelectrical PRE-POST differences. In these graphs, ellipses overlapping the origin indicate no differences between PRE and POST bioelectrical values, whereas non-centered ellipses indicate significant changes. The significance level was set at p < 0.05. Data analysis was conducted using SPSS (Version 21; Chicago, IL, USA) and BIVA software (Piccoli & Pastori, 2002) were used for data analysis.

Results

Several anthropometric, bioelectrical and CK changes were reported after the race in all groups (Table 2).

Table 2 Anthropometric, whole-body bioelectrical and CK changes evoked by the races.

		Men	Women	
		PRE	POST	% ∆	p	PRE	POST	% ∆	p	
BM (kg)									
	14 km	77.4 ± 8.1	76.7 ± 7.8	−0.8 ± 0.8	0.003	63.0 ± 9.5	62.5 ± 9.2	−0.9 ± 0.7	0.000	
	35 km	72.1 ± 8.6	70.2 ± 8.4	−2.6 ± 1.6	0.000	56.5 ± 6.0	55.7 ± 5.7	−1.3 ± 1.9	0.006	
	52 km	71.7 ± 8.4	70.1 ± 8.4	−2.2 ± 0.8	0.000	55.1 ± 3.3	54.1 ± 3.1	−1.8 ± 1.3	0.039	
R/H (Ωm)									
	14 km	266.9 ± 26.5	285.7 ± 38.0	6.8 ± 6.3	0.003	358.3 ± 38.5	364.0 ± 43.4	1.6 ± 5.1	0.214	
	35 km	268.3 ± 22.9	282.0 ± 25.0	5.1 ± 3.0	0.000	353.5 ± 41.0	365.2 ± 40.4	3.4 ± 3.1	0.000	
	52 km	275.1 ± 31.3	278.5 ± 32.7	1.2 ± 3.6	0.217	356.8 ± 37.0	359.7 ± 35.5	1.0 ± 4.6	0.702	
Xc/H (Ω/m)									
	14 km	35.3 ± 3.8	37.6 ± 3.9	6.9 ± 6.5	0.002	38.9 ± 5.3	39.1 ± 5.9	0.6 ± 7.1	0.726	
	35 km	34.4 ± 3.8	37.8 ± 4.1	9.9 ± 5.9	0.000	39.6 ± 4.9	42.2 ± 6.2	6.5 ± 6.9	0.002	
	52 km	34.2 ± 4.0	36.0 ± 4.3	5.5 ± 5.8	0.003	36.3 ± 1.9	38.7 ± 1.8	6.8 ± 2.3	0.002	
Z/H (Ω/m)									
	14 km	269.3 ± 26.7	288.2 ± 38.1	6.9 ± 6.3	0.003	360.5 ± 38.7	366.1 ± 43.7	1.5 ± 5.1	0.218	
	35 km	270.5 ± 23.1	284.5 ± 25.3	5.2 ± 3.0	0.000	355.7 ± 41.3	367.6 ± 40.7	3.4 ± 3.1	0.000	
	52 km	277.2 ± 31.4	280.8 ± 32.8	1.3 ± 3.6	0.196	358.7 ± 36.9	361.8 ± 35.4	1.0 ± 4.6	0.678	
PhA (°)									
	14 km	7.5 ± 0.5	7.6 ± 0.6	0.1 ± 4.1	0.894	6.2 ± 0.5	6.1 ± 0.6	−1.0 ± 2.8	0.179	
	35 km	7.3 ± 0.5	7.6 ± 0.5	4.5 ± 5.0	0.000	6.4 ± 0.4	6.6 ± 0.5	3.0 ± 6.4	0.098	
	52 km	7.1 ± 0.7	7.4 ± 0.6	4.1 ± 3.9	0.002	5.8 ± 0.4	6.2 ± 0.6	5.8 ± 3.8	0.033	
TBW (l)									
	14 km	48.2 ± 4.3	47.2 ± 4.5	−2.2 ± 1.6	0.000	33.7 ± 4.8	33.3 ± 4.8	−1.0 ± 1.3	0.017	
	35 km	46.0 ± 4.3	44.7 ± 4.2	−2.9 ± 1.4	0.000	31.2 ± 3.4	30.6 ± 3.2	−1.8 ± 1.5	0.000	
	52 km	45.6 ± 4.7	44.9 ± 4.7	−1.6 ± 1.3	0.001	30.5 ± 1.4	30.1 ± 1.4	−1.5 ± 2.2	0.197	
ECW (l)									
	14 km	19.1 ± 1.7	18.6 ± 1.7	−2.6 ± 2.0	0.001	14.8 ± 1.8	14.7 ± 1.9	−0.7 ± 2.0	0.221	
	35 km	18.4 ± 1.7	17.7 ± 1.6	−3.7 ± 1.7	0.000	13.9 ± 1.4	13.6 ± 1.3	−2.3 ± 1.5	0.000	
	52 km	18.3 ± 1.8	17.9 ± 1.8	−2.2 ± 1.7	0.000	13.8 ± 0.5	13.5 ± 0.4	−2.3 ± 1.7	0.042	
ICW (l)									
	14 km	29.2 ± 2.7	28.6 ± 2.8	−1.9 ± 1.4	0.000	18.9 ± 3.0	18.7 ± 3.0	−1.2 ± 1.0	0.000	
	35 km	27.6 ± 2.7	27.0 ± 2.6	−2.4 ± 1.5	0.000	17.3 ± 2.0	17.1 ± 1.9	−1.5 ± 2.2	0.009	
	52 km	27.3 ± 3.0	26.9 ± 2.9	−1.3 ± 1.2	0.001	16.7 ± 0.9	16.6 ± 1.0	−0.9 ± 2.6	0.498	
ECW/ICW ratio									
	14 km	0.66 ± 0.01	0.65 ± 0.01	−0.8 ± 1.0	0.082	0.78 ± 0.04	0.79 ± 0.04	0.4 ± 1.5	0.089	
	35 km	0.67 ± 0.02	0.66 ± 0.02	−1.4 ± 1.4	0.000	0.80 ± 0.03	0.79 ± 0.03	−0.8 ± 2.4	0.154	
	52 km	0.67 ± 0.02	0.67 ± 0.02	−0.9 ± 1.1	0.013	0.83 ± 0.02	0.82 ± 0.02	−1.4 ± 1.1	0.034	
CK (IU/l)									
	14 km	204 ± 90	378 ± 144	108.5 ± 91.9	0.002	102 ± 36 †	174 ± 72	81.6 ± 63.4	0.002	
	35 km	234 ± 246	414 ± 186	153.1 ± 123.2	0.001	138 ± 54 ‡	366 ± 318	156.3 ± 115.3	0.001	
	52 km	210 ± 102	1,020 ± 600	425.2 ± 266.9	0.001	138 ± 48 §	696 ± 390	433.5 ± 244.5	0.068	
Note:

BM, body mass; R/H, height-adjusted resistance; Xc/H, height-adjusted reactance; Z/H, height-adjusted vector length; PhA, phase angle; TBW, total body water; ICW, intracellular water; ECW, extracellular water; CK, creatine kinase; PRE, assessment before the race; POST, assessment after the race. Men: 14 km (n = 13), 35 km (n = 28), 52 km (n = 14); Women: 14 km (n = 15), 35 km (n = 17), 52 km (n = 5). †, n = 12; ‡, n = 14; §, n = 4.

Whole-body BIVA

All runners showed a decrease of BM (0.8–2.6%). WB-BIVA highlighted an increase of Z/H, especially in the 14 and 35 km groups, and PhA, especially in the 35 and 52 km groups. Paired graphs (Fig. 3) showed a migration of the vector in almost all the groups, except for women who ran 14 km. These bioelectrical changes suggest a body fluid loss, more likely at the extracellular level, as indicated by vector length and PhA increase, respectively. The trend was confirmed by the water prediction equations, showing a tendency to TBW, ECW, ICW, and ECW/ICW reduction in the majority of the cases (Table 2). PRE Z/H was negatively correlated with running time in the 35 km men group and 14 km women group (r = -0.377, p = 0.048; r = -0.751, p = 0.001; respectively). POST Z/H was negatively correlated with running time in the 14 km women group (r = -0.593, p = 0.02).

Figure 3 Paired graph.

Intra-group classic whole-body BIVA differences evoked by the race in men (A) and women (B). Solid line: 14 km; Dashed line: 32 km; Dash-dotted line: 52 km.

CK levels increased in all groups (81.6–433.5%), and it was positively correlated with distance (r = 0.521, p < 0.001 for men, and r = 0.628, p < 0.001 for women) and negatively correlated with R/H, Xc/H and Z/H in the group of men who ran 14 km only (r = -0.645, p = 0.017; r = -0.592, p = 0.033; r = -0.647, p = 0.017; respectively).

Muscle-localized BIVA

In the 11 subjects who performed ML-BIVA, the muscles with major changes were the calves (Fig. 4), with a significant increase in vector length and PhA in 35 km and, especially, 52 km runners (Table 3). Hamstrings vector length’ change showed the same pattern than calves, particularly in the 52 km group. Quadriceps changes were not significant. The changes of PhA followed the same trend as the whole-body in all cases and muscles, increasing its value (4.1–6.6%).

Figure 4 Paired graph.

Muscle-localized BIVA differences evoked by the race in 35 km group (A) and 52 km group (B). Solid line: quadriceps; Dashed line: hamstrings; Dash-dotted line, calves.

Table 3 Anthropometric and muscle-localized BIVA changes evoked by the races.

			PRE	POST	% ∆	p	
Thighs perimeter (cm)					
		35 km	52.3 ± 3.5	51.5 ± 2.6	−1.4 ± 1.8	0.148	
		52 km	53.1 ± 3.0	53.0 ± 2.7	−0.2 ± 1.3	0.753	
Calves perimeter (cm)					
		35 km	38.3 ± 1.9	37.2 ± 1.3	−2.7 ± 2.2	0.057	
		52 km	37.5 ± 0.9	37.0 ± 1.0	−1.3 ± 1.9	0.143	
Quadriceps					
	R/L (Ω/m)					
		35 km	74.5 ± 6.4	75.5 ± 10.4	1.3 ± 9.5	0.760	
		52 km	78.1 ± 9.4	75.7 ± 12.7	−3.5 ± 8.5	0.363	
	Xc/L (Ω/m)					
		35 km	21.3 ± 2.2	22.5 ± 1.3	6.4 ± 9.2	0.209	
		52 km	23.0 ± 2.8	23.0 ± 1.8	0.7 ± 10.8	0.974	
	Z/L (Ω/m)					
		35 km	77.5 ± 5.9	78.9 ± 9.7	1.7 ± 9.3	0.697	
		52 km	81.5 ± 9.3	79.1 ± 12.6	−3.2 ± 8.4	0.402	
	PhA (°)					
		35 km	16.1 ± 2.6	16.9 ± 2.9	4.8 ± 5.5	0.105	
		52 km	16.6 ± 2.3	17.2 ± 2.3	4.1 ± 5.2	0.123	
Hamstrings					
	R/L (Ω/m)					
		35 km	86.6 ± 7.2	89.7 ± 6.0	4 ± 10.6	0.487	
		52 km	92.4 ± 6.8	92.0 ± 9.3	−0.5 ± 4.9	0.838	
	Xc/L (Ω/m)					
		35 km	26.4 ± 2.6	28.7 ± 2.3	9.1 ± 8.7	0.087	
		52 km	29.5 ± 5.1	31.0 ± 4.5	5.4 ± 6.5	0.074	
	Z/L (Ω/m)					
		35 km	90.6 ± 7.1	94.3 ± 5.2	4.5 ± 10.2	0.415	
		52 km	97.1 ± 6.8	97.1 ± 9.2	0.0 ± 4.9	0.975	
	PhA (°)					
		35 km	17.0 ± 1.9	18.0 ± 2.2	5.4 ± 7.1	0.154	
		52 km	17.7 ± 3.1	18.7 ± 2.9	5.6 ± 3.0	0.001	
Calves						
	R/L (Ω/m)					
		35 km	143.2 ± 12.9	149.7 ± 12.7	4.6 ± 1.7	0.002	
		52 km	165.9 ± 13.8	176.7 ± 12.2	6.6 ± 2.5	0.001	
	Xc/L (Ω/m)					
		35 km	37.0 ± 4.9	42.3 ± 5.4	14.5 ± 6.6	0.006	
		52 km	39.9 ± 7.1	44.8 ± 7.3	12.7 ± 4.7	0.001	
	Z/L (Ω/m)					
		35 km	147.9 ± 13.7	155.6 ± 13.7	5.3 ± 2.0	0.003	
		52 km	170.7 ± 14.5	182.4 ± 12.7	6.9 ± 2.6	0.001	
	PhA (°)					
		35 km	14.5 ± 0.7	15.7 ± 0.8	4.6 ± 1.7	0.009	
		52 km	13.5 ± 1.9	14.2 ± 2.0	6.6 ± 2.5	0.001	
Note:

R/L, length segment-adjusted resistance; Xc/L, length segment-adjusted reactance; Z/L, length segment-adjusted vector length; PhA, phase angle; PRE, assessment before the race; POST, assessment after the race. 32 km (n = 5), 52 km (n = 6).

Discussion

The present study, which is the first one analyzing physiological changes by means of WB-BIVA and ML-BIVA in trail runners, highlighted body changes evoked by the demands of the races.

Whole-body changes

All groups experienced a decrease in BM and in TBW (more likely at the extracellular level), as indicated by the prediction equations, and by bioimpedance vector lengthening and PhA increase (Campa et al., 2022; Marini et al., 2020). The vector migration was similar in the three groups of runners, although some differences can be perceived. The group that lost more water was the 35 km for both men and women, and the group showing the less accentuated trend was the 14 km women group. Such differences among groups of runners could be attributed to several factors, such as the initial hydration status, the amount of water they drank during the race, the profile of the race (concentric and eccentric component) but could also be due to their physical fitness and tolerance to the race demands. It’s worth noting that similar vector migrations have been observed in two previous studies on endurance (not trail) runners (Castizo-Olier et al., 2018a; Nescolarde et al., 2020). In particular, Castizo-Olier et al. (2018a), examining experienced ultra-endurance triathletes, reported an increase of the vector length and PhA (3.8 ± 2.3%, 3.7 ± 4.9%, respectively). Nescolarde et al. (2020) showed the same trend in non-elite marathon runners (4.0% and 6.2% increase in mean for R/H and PhA, respectively). These studies performed a third assessment 48 h after the end of the race, showing that runners returned to their initial bioelectrical values after following an adequate hydration and nutritional recovery strategies. Other methods rather than BIVA, as the plasma analysis (Kim & Lee, 2015; Pradas et al., 2021), urine tests (Baiget et al., 2018), or dilution techniques (Tam, Nolte & Noakes, 2011) confirmed the tendency to dehydration in TR races.

The significant loss of TBW and BM exceeding 2% on average in some runner groups is consistent with findings in a study on 56 km ultra-distance runners using dilution techniques (Tam, Nolte & Noakes, 2011). Although it is proposed in the literature that BM losses greater than 2% impairs endurance performance (Casa, Clarkson & Roberts, 2005), it is actually unclear what are the real effects of water and BM loss on running performance (Goulet, 2012). Our study suggests that hydration status at the end of the race does not necessarily correlate with performance, with the exception of the 14 km women’s group. In this specific group, we found a positive correlation between POST Z/H and running time, indicating that those runners who finished the race in a more dehydrated state tended to perform worse. This correlation might be linked to variations in their athletic potential and their experience with effective hydration strategies, especially when compared to runners participating in longer races.

The negative correlation between PRE Z/H and running time in the 35 km men’s group was low, making it poorly relevant for our analysis. In contrast, this correlation held significance in the 14 km women’s group. Interestingly, a counterintuitive pattern emerged in these groups: higher hydration levels at the start of the race were associated with worse performance outcomes. This finding contradicts (Baiget’s et al., 2018) observation that the most dehydrated trail runners tended to be slower. It’s important to note that this relationship wasn’t consistently supported by other studies, such as Casa et al. (2010) and Noakes (2007). Noakes (2007) proposed that human physiology is naturally inclined to maintain plasma osmolality rather than focusing solely on body mass, suggesting that excessive water intake might lead to a “weight-penalty,” potentially impairing performance, as discussed previously (Gatterer et al., 2011; Noakes, 2007). Despite the fact that ECW/ICW ratio is considered an important parameter, since it is well-reported that ICW is associated with power and strength (Silva et al., 2014), it did not demonstrate a consistent relationship with performance in our study.

Muscle damage, reflected by significant CK increases, was particularly noticeable in long-distance runners. Similar results have been reported in several studies (Carmona et al., 2019; Pradas et al., 2021) and could be attributed to the continuous eccentric muscle contractions during running, leading to the release of muscle proteins like CK into the bloodstream (Brancaccio, Lippi & Maffulli, 2010). Notably, longer-distance runners might be expected to exhibit higher CK levels post-race, as observed in previous research (Temesi et al., 2021). However, it is essential to consider the superior physical fitness of longer-distance runners in comparison to other groups, which may result in their increased tolerance for elevated CK levels (Kim & Lee, 2015; Koutedakis et al., 1993). Furthermore, it is worth mentioning that the 52 km runners were preparing for an 85 km race, but the race had to be halted at the 52nd km due to a storm. Therefore, it’s plausible that they did not reach their physical limits, which could explain the lower-than-expected CK levels observed in this group.

In the present sample, significant negative correlations were detected between CK and R/H, Xc/H and Z/H in the group of men who ran 14 km only. The correlation with Xc/H was due to a decrease in cellular integrity after the race. However, a positive correlation was expected regarding R/H and Z/H, due to the crenation phenomenon that occurs in the skeletal muscle cells as a process of restoring the ECW loss (Lozano et al., 2005). In fact, a previous study with a small sample of seven subjects reported a positive correlation between R/H and CK changes (Cebrián-Ponce et al., 2022). Thus, caution is required in the interpretation of these correlations and continue researching.

Muscle-localized changes

ML-BIVA showed greater changes in the calf muscles, as indicated by large lengthening’s of the vectors and PhA increases. Hamstrings reported similar but less accentuated trends as calves, while quadriceps displayed comparatively minor changes. These bioelectrical shifts suggest a propensity towards dehydration. Based on the interpretation of whole-body PhA, it is assumed that even at the muscular level, water loss comes mainly from the extracellular compartment.

No previous studies used ML-BIVA in TR, although the technique has been applied in other sports. A study examining quadriceps before and after a high-intensity training session in rink hockey players (Cebrián-Ponce et al., 2022) detected an increase in PhA, similar to the findings in our runner sample, with subsequent normalization after 24 h. However, it is worth noting that, in contrast to our results, the hockey players experienced a decrease in Z/H along with an increase in thigh perimeter, indicative of fluid retention in the lower limb. This disparity could be attributed to the differing physiological demands of high-intensity interval efforts in hockey compared to the sustained, aerobic exertion in TR. ML-BIVA has also been employed to assess chronic adaptations to exercise in elite soccer players following 50 days of training (Mascherini, Petri & Galanti, 2015) and during the Giro d’Italia (Cebrián-Ponce et al., 2023). In alignment with our findings, the most significant changes were observed in the calf muscles.

Despite the limited existing studies on ML-BIVA in the realm of sports, particularly in TR, muscle changes have been evaluated using various methods. Multiple studies have reported reductions in isometric maximum voluntary contractions (IMVC) of knee extensors and plantar flexors following TR races (Fourchet et al., 2012; Koral et al., 2022; Pastor et al., 2022; Temesi et al., 2021), with more pronounced effects observed in longer-distance events (Temesi et al., 2021). For instance, Koral et al. (2022) observed reductions of 16% in knee extensor IMVC and 13% in plantar flexor IMVC in TR races shorter than 60 km, which further increased to 29% and 26%, respectively, in races longer than 100 km. Moreover, Baiget et al. (2018) and Pradas et al. (2021) documented significant decrements in various vertical jump tests (rebound jump; squat jump and countermovement jump; respectively). These tests serve as indicators of lower-body musculature functionality and force-generating capacity.

As evident from the literature, TR races induce peripheral fatigue, particularly in the lower limbs, resulting in reduced muscle function and performance. The calf muscles, integral to the plantar flexors, undergo significant strain during mountain running, leading to the expected erosion observed at the end of these races (Fourchet et al., 2012). The irregular terrain and high eccentric demands likely contribute to heightened activation and subsequent fatigue in posterior muscles such as the hamstrings and calves (Roca, 2019), accompanied by local fluid loss. This body of knowledge (Baiget et al., 2018; Fourchet et al., 2012; Koral et al., 2022; Pastor et al., 2022; Pradas et al., 2021; Roca, 2019; Temesi et al., 2021) suggests that BIVA is sensitive to the neuromuscular fatigue induced by TR races, particularly in the calf muscles.

Limitations and strengths

The present study has its limitations. Firstly, food and water intake during the races was not controlled. Secondly, the sample size for ML-BIVA was relatively small, comprising only male participants. Thirdly, data on POST CK levels were missing for seven participants. Fourthly, the wide variation in participant’s exercise experience, ranging from non-trained individuals to international-level athletes, may introduce potential confounding factors. Finally, the dilution technique, which typically serves as the reference method for assessing body hydration, could not be applied.

With regard to the strengths of the study, this is the first study analyzing BIVA in TR and ML-BIVA in endurance sports. Also, the study design allowed to compare the physiological responses among athletes practicing different profiles of races.

Future perspectives

In various endurance sports, continuous health and performance monitoring is essential and often necessitates mobile laboratories for swift, real-time assessments. In this context, the utilization of simple and reliable methods, such as bioimpedance monitoring, can offer valuable insights into an athlete’s hydration status, which is pivotal for their well-being and performance.

The results presented encourage further investigation in different lines of research and applications: (a) testing BIVA as a method to assess body fluid changes associated with different exercise modalities; (b) investigating the relationship between vector migration and muscle fatigue and damage, both on a whole-body and localized level; (c) extending the application of ML-BIVA to other groups and female participants. The outcomes of these investigations could equip coaches with non-invasive and straightforward tools to gain essential information about their athletes.

Conclusions

This study underscored the notable impact of trail races on both whole-body and localized parameters, with medium and long-distance runners experiencing more pronounced alterations. Anthropometrical and bioelectrical changes indicated a loss of body mass and body fluids, especially at the extracellular level. Muscle-localized BIVA showed that the major changes were at the calf level. Interestingly, our findings did not establish a straightforward link between hydration status and endurance performance, highlighting the complexity of these interactions. Muscle damage, indicated by CK changes, increased with distance.

Our study reinforces the utility of BIVA as a cost-effective, non-invasive, and efficient tool for monitoring dynamic shifts in body fluids and cellular properties. While ML-BIVA in sports remains an emerging field, our results suggest that this method may serve as a sensitive indicator of peripheral fatigue induced by high-intensity training or competitive endeavours. Further exploration in these areas could provide valuable insights into optimizing athlete health and performance.

We are most grateful to all the volunteers.

Additional Information and Declarations

Competing Interests

Author Contributions

Human Ethics

Data Availability

The authors declare that they have no competing interests.

Álex Cebrián-Ponce analyzed the data, prepared figures and/or tables, authored or reviewed drafts of the article, and approved the final draft.

Elisabetta Marini analyzed the data, prepared figures and/or tables, authored or reviewed drafts of the article, and approved the final draft.

Silvia Stagi analyzed the data, prepared figures and/or tables, authored or reviewed drafts of the article, and approved the final draft.

Jorge Castizo-Olier conceived and designed the experiments, authored or reviewed drafts of the article, and approved the final draft.

Marta Carrasco-Marginet conceived and designed the experiments, authored or reviewed drafts of the article, and approved the final draft.

Manuel Vicente Garnacho-Castaño performed the experiments, authored or reviewed drafts of the article, and approved the final draft.

Zeasseska Noriega performed the experiments, authored or reviewed drafts of the article, and approved the final draft.

Javier Espasa-Labrador performed the experiments, authored or reviewed drafts of the article, and approved the final draft.

Alfredo Irurtia conceived and designed the experiments, authored or reviewed drafts of the article, and approved the final draft.

The following information was supplied relating to ethical approvals (i.e., approving body and any reference numbers):

The research was approved by the Ethics Committee of the Catalan Sports Council (0099 S/690/2013).

The following information was supplied regarding data availability:

The data is available at Zenodo: Álex Cebrián Ponce, Elisabetta Marini, Silvia Stagi, Jorge Castizo Olier, Marta Carrasco Marginet, Manuel V Garnacho-Castaño, Zeasseska Noriega, Javier Espasa Labrador, & Alfredo Irurtia. (2023). Body fluids and muscle changes in trail runners of various distances [Data set]. Zenodo. https://doi.org/10.5281/zenodo.8342016.

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
