# Peer review of "Body fluids and muscle changes in trail runners of various distances"

_PeerJ, doi:10.7717/peerj.16563_

## Round 0.1 · original submission · Major Revisions

· Academic Editor

Major Revisions

Dear Co-Authors:

Please attend to the reviewer's comments. Some questions must be changed to be considered for publication in PeerJ Journal. The manuscript is fascinating but could be improved.

Thank you for your time and for thinking in PeerJ

Dr. Manuel Jiménez

Reviewer 1 ·

Basic reporting

Abstract Clarity:

Consider revising the abstract for enhanced clarity and succinctness. Ensure that the key findings and the significance of the study are clearly articulated in a manner accessible to a broad readership.
Terminology and Symbols:

Ensure consistency in the use of symbols and terminology throughout the manuscript. For instance, the use of “û” and “diûerent” in the text may be typographical errors and should be corrected.
Methodology Clarification:

Provide additional details about the bioelectrical impedance vector analysis (BIVA) methodology to enhance the reproducibility of the study. Specifically, elucidate on the parameters measured and the rationale behind choosing WB-BIVA and ML-BIVA.
Statistical Analysis:

Elaborate on the statistical methods used to derive the p-values mentioned in the results. Ensure that the statistical tests used are appropriate for the data distribution and sample size.
Result Presentation:

Consider presenting the results in a more structured manner, possibly utilizing subheadings to distinguish between findings related to different race distances and gender. This could enhance the readability and comprehension of the findings.
Discussion and Conclusion:

Ensure that the discussion comprehensively interprets the findings and situates them within the broader scientific context. The conclusion should succinctly encapsulate the key findings and their potential implications.
Figures and Tables:

Ensure that all figures and tables are clearly labeled and referenced within the text. Additionally, consider whether additional visual representations of the data might enhance the reader’s understanding of the findings.
References:

Ensure that all references are formatted consistently and adhere to the PeerJ citation style. Verify that all citations within the text are listed in the reference section and vice versa.
Ethical Considerations:

Confirm that all ethical considerations, including informed consent and approval from an institutional review board, are explicitly stated and adhered to in the methodology.
Language and Grammar:

Ensure that the manuscript is free from grammatical errors and employs clear and concise language throughout. Consider utilizing a professional editing service if necessary.

Experimental design

The study under discussion aims to explore the physiological alterations, particularly in body fluids and muscle changes, experienced by trail runners across various race distances. The experimental design is crafted to capture both anthropometric and bioelectrical changes, alongside creatine kinase (CK) measurements, to gauge muscle damage and potential shifts in hydration status

Validity of the findings

The study investigating body fluids and muscle changes in trail runners across various distances (14 km, 35 km, and 52 km) through bioelectrical impedance vector analysis (BIVA) and creatine kinase (CK) measurements provides insightful data regarding post-race dehydration and muscle damage tendencies. While the inclusion of diverse race distances and both genders enhances the generalizability of the findings, the validity of the findings warrants a thorough examination in aspects like the control of external variables (e.g., weather, hydration strategies), the reliability and specificity of BIVA and CK in this particular context, and the representativeness and homogeneity of the sample concerning physical fitness and training regimes. Future research might benefit from considering these aspects and potentially incorporating additional markers and methodologies to validate and expand upon these findings in the realm of trail running and endurance sports.

·

Basic reporting

No comment

Experimental design

In the section on methods, although the sample is large, I consider that the information provided, with respect to the subjects, is not sufficient. It would be advisable to add the years of experience in competition and weekly training hours, as has been done in similar research, such as Carmona et al., 2019 (cited in this article). Since the only information provided is "The runners were asked about their weekly training volume, and they revealed a great heterogeneity among them, as usual among TR runners." If you do not have this data, it should appear in limitations

Carmona, G., Roca, E., Guerrero, M., Cussó, R., Bàrcena, C., Mateu, M., & Cadefau, J. A. (2019). Fibre-type-specific and Mitochondrial Biomarkers of Muscle Damage after Mountain Races. International journal of sports medicine, 40(4), 253-262. https://doi.org/10.1055/a-0808-4692

Validity of the findings

Regarding the results section, very useful information is obtained. I find very interesting the application of correlations between bioelectrical and CK delta percent changes, and also between PRE and POST bioelectrical parameters and running time. But I do not understand why only 3 correlations are referred to in the results. I would be grateful if you could tell me which were all the correlations, as well as clarify me why it is due to take into account a correlation of -0.377 between PRE Z/H and the male group of running time in the 35 km, since it is a low correlation, and despite this in the discussion you can read:
"The negative correlation between PRE Z/H and running time in the 35 km men group and 14 km women group of our study indicated that in these groups the more hydrated at the beginning of the race showed the worse performance."
With a correlation it would be more correct to put "may suggest", and if it is so low it should not be included. I don't know if there has been an error and the correlation is actually higher. I would appreciate a clarification on this.

Additional comments

No coment

---

## Round 0.2 · accepted · Accept

· Academic Editor

Accept

Dear Authors:

Thank you for considering PeerJ to publish your manuscript. I am pleased to announce that PeerJ Journal has just accepted your article.

Congratulations